# A global lipid map reveals host dependency factors conserved across SARS-CoV-2 variants

Scotland E. Farley[1,2], Jennifer E. Kyle [3], Hans C. Leier[1], Lisa M. Bramer [3], Jules B. Weinstein [1], Timothy A. Bates [1], Joon-Yong Lee [3], Thomas O. Metz [3], Carsten Schultz[2] & Fikadu G. Tafesse [1✉]

A comprehensive understanding of host dependency factors for SARS-CoV-2 remains elusive. Here, we map alterations in host lipids following SARS-CoV-2 infection using non-targeted lipidomics. We find that SARS-CoV-2 rewires host lipid metabolism, significantly altering hundreds of lipid species to effectively establish infection. We correlate these changes with viral protein activity by transfecting human cells with each viral protein and performing lipidomics. We find that lipid droplet plasticity is a key feature of infection and that viral propagation can be blocked by small-molecule glycerolipid biosynthesis inhibitors. We find that this inhibition was effective against the main variants of concern (alpha, beta, gamma, and delta), indicating that glycerolipid biosynthesis is a conserved host dependency factor that supports this evolving virus.

[1] Department of Molecular Microbiology & Immunology, Oregon Health & Science University, Portland, OR, USA. [2] Department of Chemical Physiology and Biochemistry, Oregon Health & Science University, Portland, OR, USA. [3] Biological Sciences Division, Earth and Biological Sciences Directorate, Pacific Northwest National Laboratory (PNNL), Richland, WA, USA. ✉email: tafesse@ohsu.edu

SARS-CoV-2 interacts with host membranes at every stage of its life cycle. It directly crosses the plasma membrane to enter the cell, replicates inside host-derived membrane compartments, acquires its envelope from the host, and traffics through the Golgi and lysosome to exit the cell. All viruses, by their nature, are wholly dependent on host pathways to meet their metabolic, structural, and trafficking needs, and to be effective, they must modulate these host pathways in some way. One dramatic example of this is the way in which SARS-CoV-2 re-engineers the host internal membranes into double-membraned vesicles (DMVs) and regions of convoluted membrane (CM) to facilitate its replication[1,2]. This general pattern of membrane rearrangements is common among (+)-stranded RNA viruses[3–5], although the specific structures vary by species. In flaviviruses such as Zika virus[6] and dengue virus[7], these large-scale membrane alterations are accompanied by vast and varied changes at the molecular lipid level.

There are many preliminary lines of evidence suggesting that manipulation of host lipids may be a fundamental feature of SARS-CoV-2 infection. Several lipids and lipid-associated proteins have been identified as biomarkers of infection, including VLDL and HDL particles[8], steroid hormones and various apolipoproteins[9], while both elevated triacylglycerol (TAG)[10] and polyunsaturated free fatty acids[11] have been implicated as markers of severe disease outcomes. Furthermore, patients with a high BMI, diabetes, or hypertension are at a higher risk of developing severe disease[12]. These observations indicate systemic changes in lipid metabolism at an organismal level, but it is still unknown how the virus alters the host lipid metabolism at a cellular level, and how these changes support the viral life cycle.

Here, we show that SARS-CoV-2s reprograms host lipid biosynthesis and depends on specific host metabolic pathways to survive and replicate effectively. To obtain a comprehensive understanding of how SARS-CoV-2 remodels the cellular lipid composition, we perform a detailed lipid survey of both infected cells and cells ectopically expressing individual SARS-CoV-2 proteins, detailing massive changes in host lipid composition as a result of infection and as a result of the activity of specific viral proteins, especially among neutral lipids. Based on our initial results showing a strong induction of neutral lipids, we examine lipid droplet flux during infection and observe dramatic proliferation of lipid droplets, and demonstrate the requirements for specific host lipids using small-molecule inhibitors of glycerolipid biosynthesis in multiple strains of SARS-CoV-2.

## Results

**Lipidomics of SARS-CoV-2 infected human cells.** We performed global lipidomic profiling in two cell lines after 24 h of infection (Fig. 1A and Supplementary Fig. 1): HEK293T cells overexpressing the ACE2 protein (HEK293T-ACE2), in order to be able to correlate the results with viral-protein-transfected cells, and the more physiologically relevant cell type A549-ACE2 cells, to model the cells affected in natural infection. Each condition was repeated in biological quintuplicate. Total cellular lipids were extracted following the method of Bligh-Dyer[13] and analyzed by liquid chromatograph electrospray ionization tandem mass spectrometry (LC-ESI-MS/MS). The abundances of the identified lipids were normalized by comparison to internal standards for quantitative analysis. In HEK293T-ACE2 cells, we identified 514 unique lipids spanning the glycerolipid, phospholipid, sphingolipid, and acylcarnitine categories (Supplementary Data 1). Of these, 409 (79.6%) were statistically altered between SARS-CoV-2 and mock infection (Benjamini–Hochberg adjusted $P < 0.05$, analysis of variance [ANOVA] test), changing between 2- and 64-

fold in response to infection. In A549-ACE2 cells, we identified 443 unique lipids spanning the same categories as above (Supplementary Data 2). Of these, 227 were statistically altered between SARS-CoV-2 and mock infection (Benjamini-Hochberg adjusted $P < 0.05$, analysis of variance [ANOVA] test), changing between 1.1- and 19.7-fold in response to infection. Principal component analysis (PCA) of these observations confirmed that infection status accounted for most of the changes (Fig. 1B and Supplementary Fig. 2), with the five infected samples and the five mock samples falling into two distinct groups.

We then examined how these changes in host lipid composition broke down based on class and acyl chain. In both cell lines, TAG species increase dramatically, as do ceramides (Fig. 1C, D). Examining the nature of the individual lipid species that changed in more detail (Fig. 1C), we observed that the TAG species change based on their fatty acid composition. TAG species that bear polyunsaturated fatty acid (PUFA) chains were increased an average of 8-fold more than saturated or monounsaturated species in 293T-ACE2 cells, and an average of 2-fold more than saturated or monounsaturated species in A549-ACE2 cells. This trend was also observed in phospholipids: saturated phospholipids (phosphatidylcholine, PC; phosphatidylethanolamine, PE; phosphatidylglycerol, PG; phosphatidylinositol PI) almost universally decreased, while many polyunsaturated species increased, notably P-PC (phosphatidylcholine, plasmalogen-linked) (2.7-fold, HEK293T-ACE2; 1.5-fold, A549-ACE2), PC (1.5-fold, HEK293T-ACE2; 1.4-fold, A549-ACE2) and PG (1.7-fold, HEK293T-ACE2, 1.3-fold, A549-ACE2). There are some differences in the lipid remodeling between the two cell lines; in particular, the effect on PI species, with PI generally decreasing in HEK293T-ACE2 cells, and increasing in A549-ACE2 cells. Cardiolipin decreases in HEK293T-ACE2 cells but does not change in A549-ACE2 cells; cholesterol esters increase in A549-ACE2 cells but not in HEK293T-ACE2 cells. While some differences in lipid metabolism between cell lines are not unexpected, the striking similarities in many lipid species suggest that SARS-CoV-2 has defined lipid requirements that it engineers in different cell types it infects.

**Lipidomics of human cells ectopically expressing SARS-CoV-2 proteins.** The genome of SARS-CoV-2 encodes 29 individual proteins (Fig. 2A). Some of these proteins have been directly studied in SARS-CoV-2, but the roles of most of them must be extrapolated by comparison with the proteins of SARS-CoV, which are better studied (Fig. 2B). Several SARS-CoV proteins directly manipulate cellular membranes—nsp3, nsp4, and nsp6 together are known to induce DMVs[14] and CMs[15,16] characteristic of coronavirus infection[17,18], and orf6[19] also has a dramatic membrane-remodeling phenotype. Some proteins of SARS-CoV-2—nsp1, nsp8, nsp9, nsp16—have direct effects on mRNA splicing, or protein expression and membrane integration[20]. Many proteins of both viruses contain transmembrane domains (nsp2[21], orf7a[22], orf7b[23], orf3a[24,25]) or lipid binding pockets (orf9b[26]) of unknown function, and many others, including nsp3[27], nsp6[28], orf3a[29,30], orf6[31], and orf7a[32]—mediate cell distress pathways such as apoptosis, autophagy, and the unfolded protein response (UPR), which are all known to alter cellular lipid composition[33–35] (Fig. 2B).

To assess how each SARS-CoV-2 protein affects host lipid metabolism, we performed untargeted lipidomic profiling of cells transfected with each viral protein, expressed in the PLVX vector with a C-terminal Strep tag. We optimized the expression of each protein in HEK-293T cells, measuring transfection efficiency by immunofluorescence of the Strep tag (Supplementary Fig. 2). In order to make meaningful comparisons between these conditions,

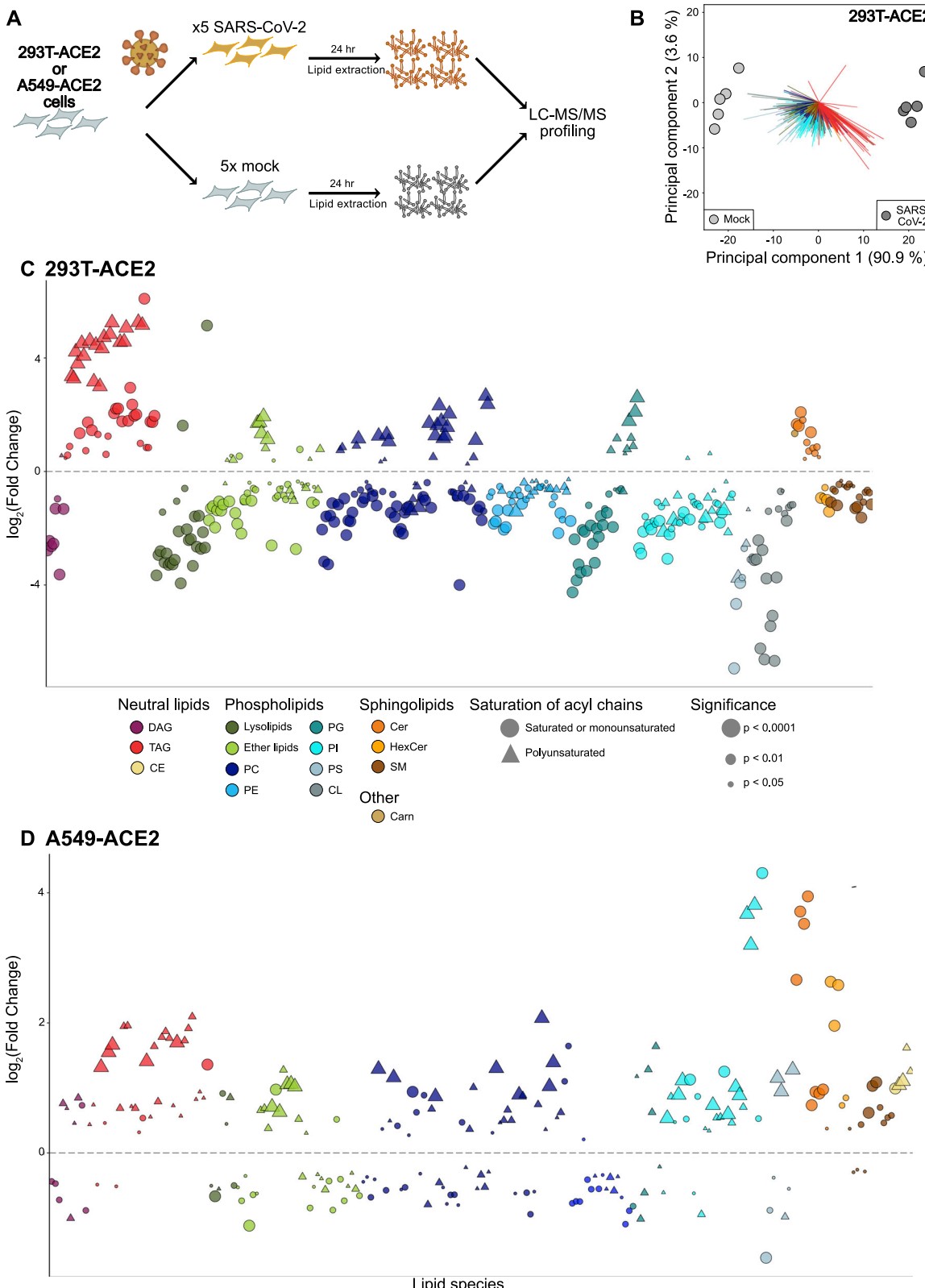

high transfection efficiency was required (>70%). Despite our efforts, this level of efficiency was not achieved for five proteins (nsp3, nsp14, nsp15, nsp16, orf3b); therefore, we continued on with the remaining 24. Each viral protein, as well as the PLVX empty vector, was used to transfect 6-cm dishes of HEK293T cells in biological quintuplicate. After 48 h of transfection, total cellular lipids were extracted following the method of Bligh-Dyer[13] and

analyzed by liquid chromatography-electrospray ionization tandem mass spectrometry (LC-ESI-MS/MS) (Fig. 2C). The abundances of the identified lipids were normalized by comparison to internal standards for quantitative analysis. In total, we identified 396 unique lipids spanning the glycerolipid, phospholipid, sphingolipid, and acylcarnitine categories (Supplementary Data 3). Of these, 317 (80%) were significantly changed in at

**Fig. 1 SARS-CoV-2 alters the lipid composition of its host cells. A** Lipidomics study design. **B** Principal component analysis of 293T-ACE2 cells infected with SARS-CoV-2 or mock-infected ($n = 5$ biological replicates; each point represents one biological replicate). **C** Individual lipid species characterized by abundance in SARS-CoV-2 infection relative to mock in HEK293T-ACE2 cells (data points are means from five biological replicates; each data point represents a lipid species). Only significantly changed ($P < 0.05$, one-way ANOVA, with Benjamini–Hochmini adjustment for multiple comparisons) lipids are shown. $\text{Log}_2$(Fold Change) relative to mock infection is shown on the x-axis. Individual lipid species are colored by the class of lipid that they belong to. DAG diacylglycerol; TAG triacylglycerol; PC phosphatidylcholine; PE phosphatidylethanolamine; PG phosphatidylglycerol; PI phosphatidylinositol; PS phosphatidylserine; CL cardiolipin; Cer ceramide; HexCer hexosylceramide; SM sphingomyelin; Carn acylcarnitine; CE cholesterol ester. **D** Individual lipid species characterized by abundance in SARS-CoV-2 infection relative to mock in A549-ACE2 cells (data points are means from 5 biological replicates; each data point represents a lipid species). Only significantly changed ($P < 0.05$, one-way ANOVA, with Benjamini–Hochmini adjustment for multiple comparisons). Same colors and abbreviations as in (**C**) apply. Source data are provided in Supplementary Data 1 (HEK293T-ACE2 cells) and Supplementary Data 2 (A549-ACE2 cells).

least one transfection (Benjamini–Hochberg adjusted $P < 0.05$, ANOVA).

For the samples transfected with viral proteins, we performed an EASE (Expression Analysis Systematic Explorer) score enrichment test of statistically significant lipids using Lipid Mini-On. Lipid Mini-On performs enrichment analyses of lipidomics data using a text-mining process that bins individual lipid names into multiple lipid ontology groups based on their classification and other characteristics, such as chain length and number of double bonds. Using Lipid Mini-On we found that the most common enrichments that were increased with the viral-protein transfections were PIs, (elevated in 21 of 28 transfections), diacylglycerols (DAGs) (12 transfections), and ether-linked lipids, in particular vinyl-ether phosphatidylcholines (O-PC) (10-12 transfections), Cer (10 transfections), and TAG (6 transfections). Enrichments that were found to be decreased were Lyso-PC (decreased in 21 transfections), CLs (12 transfections), which almost universally decrease in abundance, and TAGs (decreased in 14 transfections) (summarized in Fig. 2, D–F; fold changes for all significant lipids are shown in Supplementary Fig. 3). The 24 viral proteins studied show a wide variety of lipid alterations, suggesting that SARS-CoV-2 influences host lipid metabolism in diverse ways through multiple molecular mechanisms.

**Correlating live virus and viral-protein lipidomic phenotypes.** With three substantial datasets of virus-induced lipid changes, we sought to link the changes observed in live virus infection to the action of specific viral proteins. First, we performed unsupervised clustering of the normalized lipid species observed in the protein-transfected dataset by t-SNE (Fig. 3A). While most phospholipids did not cluster substantially, TAG, in particular, formed distinct clusters, and, in an echo of the live virus phenotype, saturated species and polyunsaturated species clustered separately. Of note, two other molecular features of infection—Cer and CL—also sorted into distinct clusters.

We compared the two live virus infection datasets to select lipid phenotypes that were common results of infection between the two cell lines. These are significant changes in TAG, Cer, and phospholipids bearing polyunsaturated fatty acyl chains, with a decrease in DAG and saturated phospholipids (Fig. 3B). In order to assess each viral protein for its ability to produce these changes, the average fold change for each of these classes was calculated for each condition (Fig. 3C). Once again we saw that the virus has multiple proteins that influence remodeling of the lipid environment of its host cells, suggesting a distinct role for each viral protein. Each feature of infection was recapitulated by at least one protein, and different proteins appear to be responsible for different aspects of the live virus lipid phenotype.

In particular, TAG increase was recapitulated by five proteins (orf6, nsp13, nsp5, orf9c, nsp1). Cer increase was recapitulated by six as well (nsp6, orf6, nsp5, orf9c, orf3a, and orf7a), and polyunsaturated PC (both ether- and ester-linked) increase was recapitulated by four (orf6, orf9c, orf9b, and E). Of note, orf6 and orf9c recapitulated all three of these distinctive alterations, and also recapitulated the most individual phenotypes of any protein.

**Lipid droplet dynamics in SARS-CoV-2 infection.** TAG is the most significantly and the most substantially increased lipid in response to viral infection. TAG is produced through the acylation of DAG by DGAT1 or DGAT2, where it is then sequestered in lipid droplets that can be accessed as a source of fatty acids. TAG breakdown is the result of several lipases that remove an acyl chain to produce DAG (Fig. 4A). Lipid droplets (LDs) are the cellular reservoir for TAGs, and have well-established roles in the life cycles of other viruses. Hepatitis C virus (HCV) and rotaviruses both cause LDs to accumulate during infection, and HCV uses LDs as the site of viral assembly while rotavirus replication occurs in close proximity to lipid droplets[36,37]. Dengue virus, meanwhile, consumes host lipid droplets and appears to use them as a source for beta-oxidation[38].

We sought to understand how the abundance and morphology of host lipid droplets changes during the course of SARS-CoV-2 infection, and whether they are associated with virus-induced membrane structures. We chose BODIPY 493/503, a bright, hydrophobic dye, to mark the lipid droplets, a well-established method[39]. We also used an anti-dsRNA antibody to mark the sites of viral replication; dsRNA is an intermediate in the synthesis of the virus's RNA genome, and has been shown to localize to DMVs[14]. We visualized both of these markers 8 h, 24 h, or 48 h post-infection in HEK-293T cells and then stained with BODIPY 493/503 to mark lipid droplets and an anti-dsRNA antibody to mark the site of viral replication (Fig. 4B). We see a clear increase in the number and size of lipid droplets in a time-dependent manner over the course of infection, quantified in Fig. 4C, D. Lipid droplets per cell increase from zero at 8hpi, to an average of 6.7 at 24 hpi, to an average of 21.5 at 48 hpi. Lipid droplet area increases from zero pixels per droplet at 8 hpi, to an average of 177 pixels per droplet at 24 hpi, to an average of 400 pixels per droplet at 48 hpi. However, there does not appear to be any colocalization of the lipid droplets and dsRNA, suggesting that the virus is not using lipid droplets directly as a platform for replication (Fig. 4E).

To further validate these observations, similar experiments were performed in the human epithelial Caco2 cell line. Here, a slight increase in lipid droplet number was observed, from an average of 8 lipid droplets per cell at 8 and 24 hpi, to an average of 15.9 lipid droplets per cell at 48 hpi, although the increase was not significant. Lipid droplet area, however, did significantly increase throughout the course of infection, to a similar degree as in HEK293T-ACE2 cells, from an average of 136.5 pixels per droplet at 8 hpi to an average of 192.5 pixels per droplet at 24hpi to an average of 431.1 pixels per droplet at 48 hpi (Fig. 4F, G). Once again, colocalization with dsRNA was not observed (Fig. 4H).

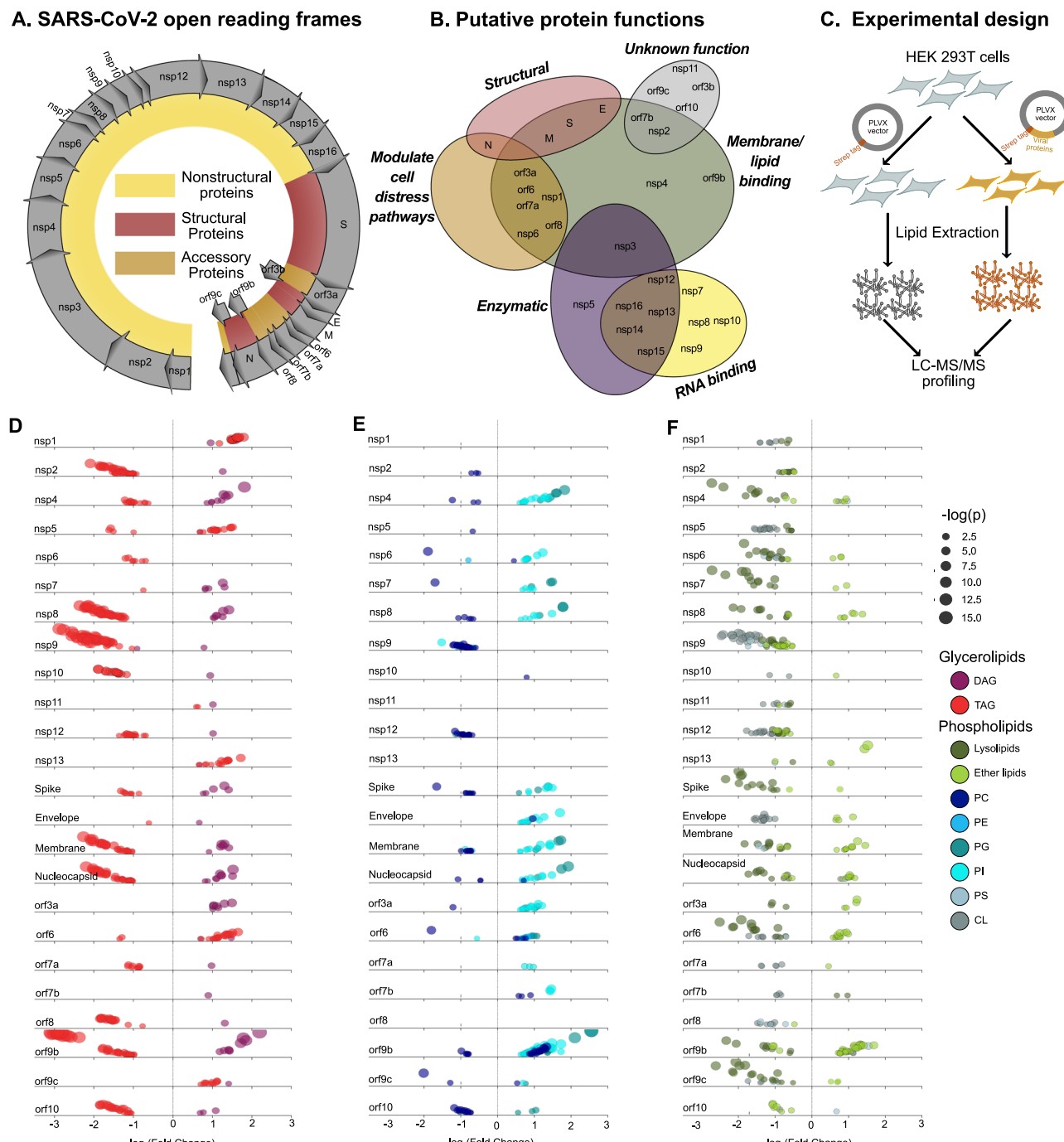

**Fig. 2 Ectopic expression of SARS-CoV-2 proteins modulates host lipid metabolism. A** Open reading frames of the SARS-CoV-2 genome. **B** Putative functions of SARS-CoV-2 proteins, based on early studies and sequence similarity to proteins from SARS-CoV. **C** Lipidomics study design. **D**–**F** Individual lipid species characterized by class and family (data points are means from five biological replicates; each data point represents a lipid species). Only significantly changed ($P < 0.05$, one-way ANOVA, Benjamini–Hochmini adjusted for multiple comparison) lipids are shown. Log$_2$(Fold Change) relative to empty vector is shown on the $x$-axis. Individual lipid species are colored by the class of lipid that they belong to. Abbreviations same as Fig. 1. Source data are provided in Supplementary Data 3.

Since our lipidomics experiments indicated that five individual SARS-CoV-2 proteins were able to independently induce the production of TAG (nsp1, nsp5, nsp13, orf6, and orf9c), we asked whether these proteins were also able to alter the formation of lipid droplets. We transfected HEK293T cells with each of these plasmids, as well as an empty vector control, and stained them with BODIPY493/503 to visualize lipids droplets and an anti-Strep antibody to identify transfected cells (Fig. 4I, quantified in Fig. 4J). We observed a strong and significant induction of lipid droplets for four of the five proteins relative to empty vector (nsp1, nsp5, orf6, and orf9c), suggesting that these proteins play a direct role in stimulating TAG synthesis, which would naturally cause the formation of lipid droplets.

**Viral requirements for central glycerolipid metabolism.** Since levels of individual glycerolipid species as well as glycerolipid-

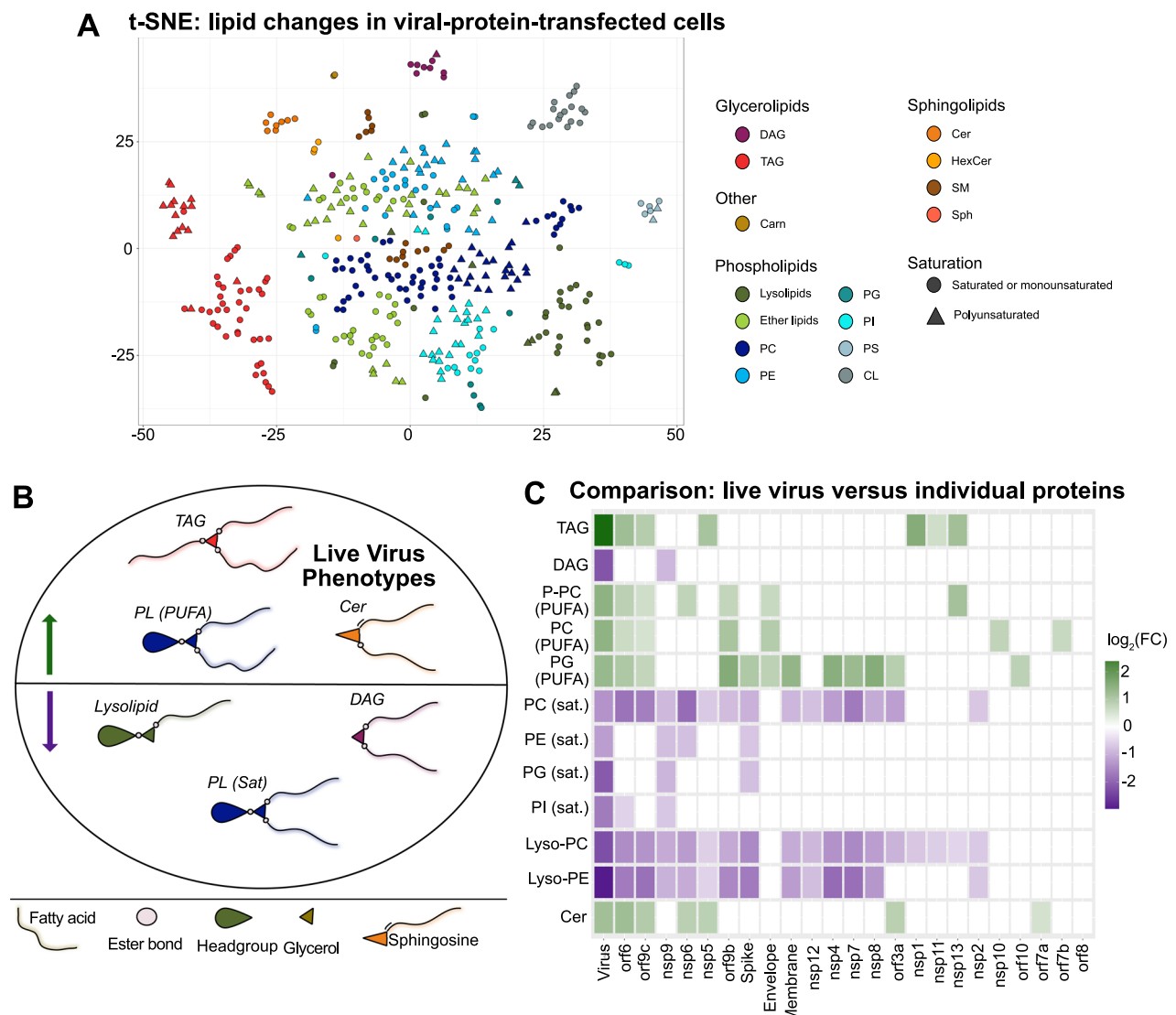

**Fig. 3 Individual SARS-CoV-2 proteins recapitulate overlapping lipid features of live infection. A** Unsupervised clustering of the normalized lipid species observed in the protein-transfected dataset by t-SNE. Abbreviations same as Fig. 1. **B** Summary of lipids altered upon infection with SARS-CoV-2 in both HEK293T-ACE2 cells and A549-ACE2 cells. Cer = ceramide; PL (PUFA) = phospholipids bearing polyunsaturated acyl chains; TAG = triacylglycerol; PL (Sat) = phospholipids bearing saturated or monounsaturated acyl chains. **C** Average fold change within each class described above in each condition, both live virus infection (HEK293T-ACE2 cells) and ectopic protein expression. Only significantly changed (P < 0.05, see Figs. 1 and 2 for descriptions of statistical tests in the live virus and transfection conditions, respectively) lipid species were used in this calculation. Source data are provided as a Source Data file.

based structures were altered by infection, we asked whether these pathways are necessary for viral proliferation. We selected an array of commercially available small molecule inhibitors of lipid synthesis, focusing on inhibitors of de novo neutral lipid synthesis as well as lipolytic enzymes of lipid recycling (Fig. 5A–F). We performed initial cytotoxicity measurements using a resazurin-based viability assay[40] (Supplementary Fig. 4) and selected a non-cytotoxic concentration of each compound to screen for inhibition of viral infection. HEK293T-ACE2 cells were treated overnight with each compound, and then infected with SARS-CoV-2. After 48 h of infection, culture supernatants were collected and the amount of infectious virus produced in the presence of each compound was quantified by focus-forming assay[41].

This screen revealed several steps of lipid biosynthesis which are essential to the production of infectious virions. De novo fatty acid synthesis appeared critical, as GSK2194069, an inhibitor of fatty acid synthase (FASN)[42], as well as Orlistat, a non-specific

lipase inhibitor and inhibitor of fatty acid synthetase FASN[43,44], an FDA-approved drug, both completely blocked viral production (Fig. 5C, D). TAG synthesis and lipolysis are both required, as PF-04620110, an inhibitor of DGAT1[45], Orlistat, and CAY10499, which is a non-specific lipase inhibitor[46,47], all blocked infection (Fig. 5E, D, B). Atglistatin[48], which specifically blocks adipose triacylglycerol lipase, partially inhibited viral production (Fig. 5A), suggesting that broad-spectrum lipase inhibition is more effective than inhibiting only one lipase. The importance of DAG production to the virus, perhaps as a precursor to TAG, is indicated by the efficacy of U-73122 (Fig. 5F), which inhibits phospholipase-C-dependent processes[49].

To directly compare the inhibitors of central glycerolipid metabolism, we designed a more detailed study to test a range of concentrations for each inhibitor. We tested a range of two-fold dilutions of each compound, and in parallel with the focus-forming assay to assess viral replication, we performed a

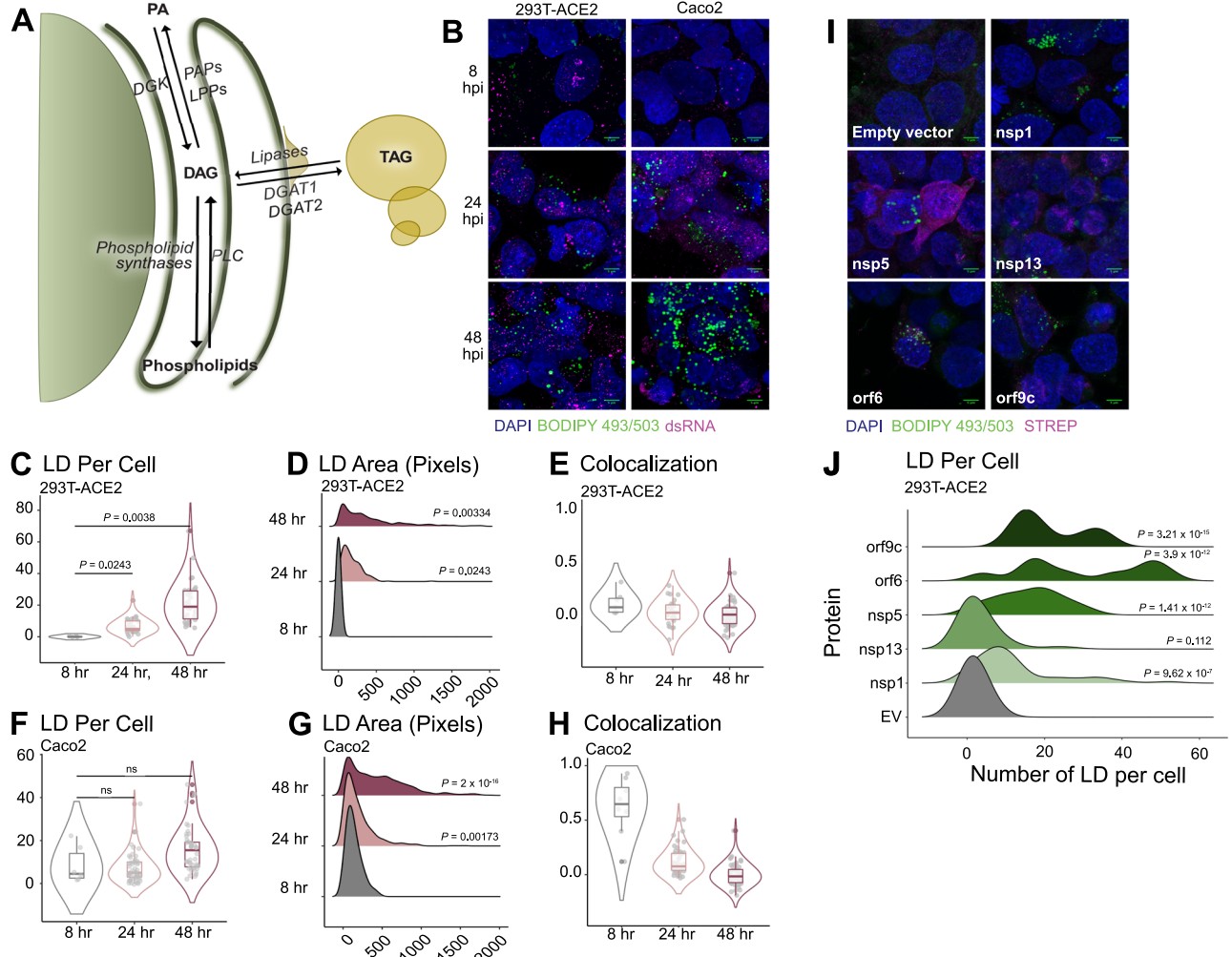

**Fig. 4 Lipid droplets are induced following SARS-CoV-2 infection and after the transfection of key viral proteins. A** Overview of central glycerolipid metabolism. PA = phosphatidic acid; PAP = phosphatidic acid phosphatase; LPP = lysophosphatidic acid phosphatase; DGK: diacylglycerol kinase; DAG = diacylglycerol; TAG = triacylglycerol; DGAT 1/2 = diacylglycerolacetyltransferase 1/2; PLC = phospholipase C. **B** 293T-ACE2 and Caco-2 cells infected with SARS-CoV-2 strain USA-WA1/2020 (MOI = 1) and fixed at the indicated timepoints. LDs and infected cells were visualized with BODIPY 493/503 and anti-dsRNA immunofluorescence, respectively. Images are representative of three independent experiments. **C, F** Number of lipid droplets per cell; each data point is a cell. *P*-values are derived from one-way ANOVA relative to the 8hpi condition. **D, G** Distribution plot of the area of each individual lipid droplet from (**C, F**), in pixels. *P*-values are derived from one-way ANOVA relative to the 8hpi condition. **E, H** Colocalization of dsRNA and BODIPY by Pearson's coefficient. Each data point is a cell. *P*-values are derived from one-way ANOVA relative to the 8hpi condition. **I** 293T cells transfected with the indicated viral proteins, 48 h after transfection. Images are representative of two independent experiments. **J** Distribution plot of the number of lipid droplets per cell in each transfection. *P*-values are derived from one-way ANOVA relative to the Empty Vector condition. The box plots are presented with the elements: center line, median; box limits, Q1 and Q3; whiskers, 1.5 x interquartile range. Outliers are also shown. Source data for all panels are provided as a Source Data file; n numbers (representing cells) for each condition can be found in the Source Data file.

resazurin-based cytotoxicity assay to verify that any deficiency in viral production was not due to impaired cell viability (Supplementary Fig. 4). The most effective inhibitor by about fifty-fold was GSK2194069 (EC$_{50}$ = 1.8 nM, HEK293T-ACE2). GSK2194069 blocks FASN, suggesting that de novo lipid synthesis is strictly required for viral survival. Orlistat followed in efficacy (EC$_{50}$ = 94 nM, HEK293T-ACE2), highlighting the importance of both fatty acid synthesis and lipolysis to the virus. The other broad-spectrum lipase inhibitor, CAY10499 (EC$_{50}$ = 157 nM, 283T-ACE2) had a similar efficacy to PF04620110 (EC$_{50}$ = 490 nM, 293T-ACE2). Atglistatin, the most specific lipase inhibitor, became cytotoxic before complete inhibition was achieved, and so an EC$_{50}$ could not be calculated; certainly it is higher than 10 μM, showing again that the virus is not dependent on the activity of one specific lipase, but rather on a certain lipid composition. Taken together, these results indicate a

profound dependence on host lipid metabolism, and in particular glycerolipid flux. The de novo synthesis of TAG is required, as is the ability to release the fatty acids sequestered in this neutral storage lipid through lipolysis.

**Glycerolipid biosynthesis as a host dependency factor.** Given that our most effective inhibitors all relate in some way to the dynamics of TAG production, we hypothesized that their efficacy is due to the virus's specific requirements for lipid droplets. We performed microscopy of cells treated with selected inhibitors at 10 μM overnight (Fig. 6A, quantified in Fig. 6B, experimental scheme in Supplementary Fig. 5). We once again observed that virus alone induced a significant increase in the number of lipid droplets per cell, from an average of 0.3 to average of 3, and further noted that in the absence of virus,

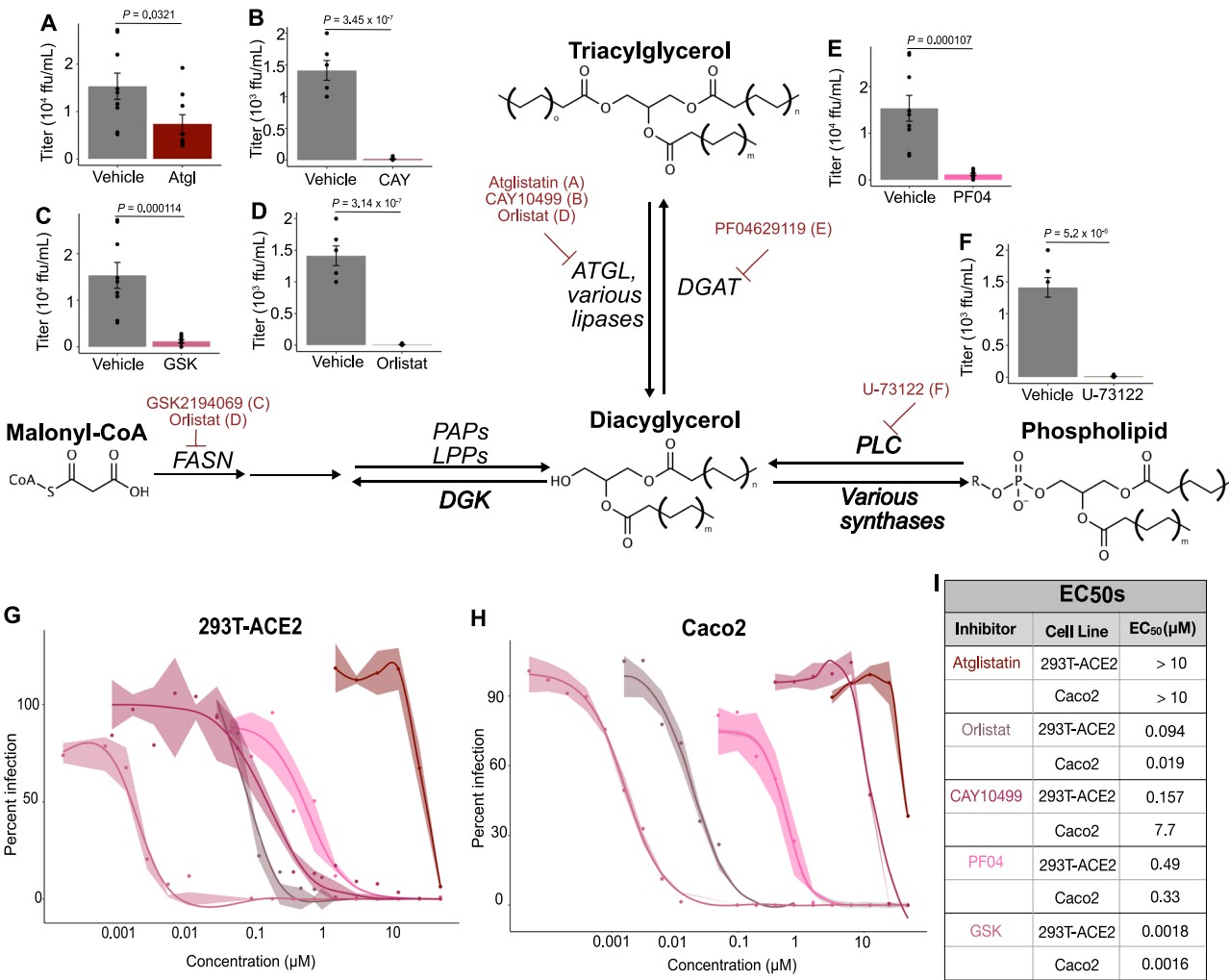

**Fig. 5 Central glycerolipid metabolism is essential for SARS-CoV-2 infection. A–F** Screen of neutral lipid biosynthesis inhibitors. HEK-293T-ACE2 cells were treated with 10 µM of each compound overnight prior to infection. Cells were infected for 48 h prior to supernatant collection. Bars represent viral titers from cells treated with the indicated inhibitors, measured by focus-forming assay. The box plots are presented with the elements: center line, median; box limits, Q1 and Q3; whiskers, 1.5 x interquartile range, from three independent experiments; individual data points are also shown, representing biological replicates (*n* = 9). *P*-values are derived one-way ANOVA. FASN = fatty acid synthase; PAP = phosphatidic acid phosphatase; LPP = lipid phosphate phosphatase; DGK = diacylglycerol kinase; ATGL = adipose triacylglycerol lipase; DGAT = diacylglycerolacetyltransferase; PLC = phospholipase C **G** EC$_{50}$ curves for selected inhibitors in 293T-ACE2 cells. HEK-293T-ACE2 cells were treated with 2-fold dilutions of each compound overnight prior to infection. Cells were infected for 48 h prior to supernatant collection and focus-forming assay. Percent infection is calculated as [Titer(inhibitor) /Titer(vehicle)]*100. Data are from three independent experiments. Error band is SE; the measure of center for the error band is the mean. Curve fits are calculated using a nonlinear curve fit to the Hill equation: Response = (Max Response)/(1 + [EC$_{50}$/Concentration]^*n*), where the max response is defined as 100% inhibition. **H** EC$_{50}$ curves for selected inhibitors in Caco2 cells. Experiment and analysis same as described in (**G**). **I** EC$_{50}$ values from the curves in G and H. EC$_{50}$ values are calculated from the curve fit described above. Source data are provided as a Source Data file.

none of the inhibitors had an effect on lipid droplet numbers. In the presence of virus, GSK2194069 treatment did not prevent a statistically significant increase in lipid droplet numbers, while PF04620110 did, suggesting that DGAT1 is essential for virus-induced lipid droplet production. Orlistat, meanwhile, resulted in an increase in lipid droplet numbers relative to vehicle treatment during infection, from an average of 3 to an average of 7.5. These results underscore the specificity of SARS-CoV-2's requirements for lipid droplets: while SARS-CoV-2 infection results in an overall increase in the lipid droplets in each infected cell, both TAG synthesis and lipolysis are required to support the production of infectious virions. Furthermore, simply increasing the number of lipid droplets does not support replication: pure accumulation of TAG resulting from the inhibition of lipolysis is as detrimental to infection as preventing its synthesis.

SARS-CoV-2 interacts with host lipids at every stage of its life cycle. To rule out the possibility that glycerolipid metabolism is necessary for the initial attachment and endocytosis of the virus, we performed an entry assay using spike-pseudotyped lentivirus. For this experiment, lentiviruses were generated that display the SARS-CoV-2 spike protein and carry a GFP reporter; lentiviruses coated instead with the VSV G protein were used as a control. Successfully infected cells express GFP, and quantitative microscopy was used to assess infection (Supplementary Fig. 5B). HEK293T-ACE2 cells were treated overnight with selected inhibitors of glycerolipid biosynthesis and then infected with either of these two lentivirus constructs. We did not observe a significant reduction in viral entry in the presence of any of the inhibitors tested, suggesting that the virus depends upon this lipid biosynthetic pathway to facilitate the intracellular stages of its life cycle (Fig. 6D).

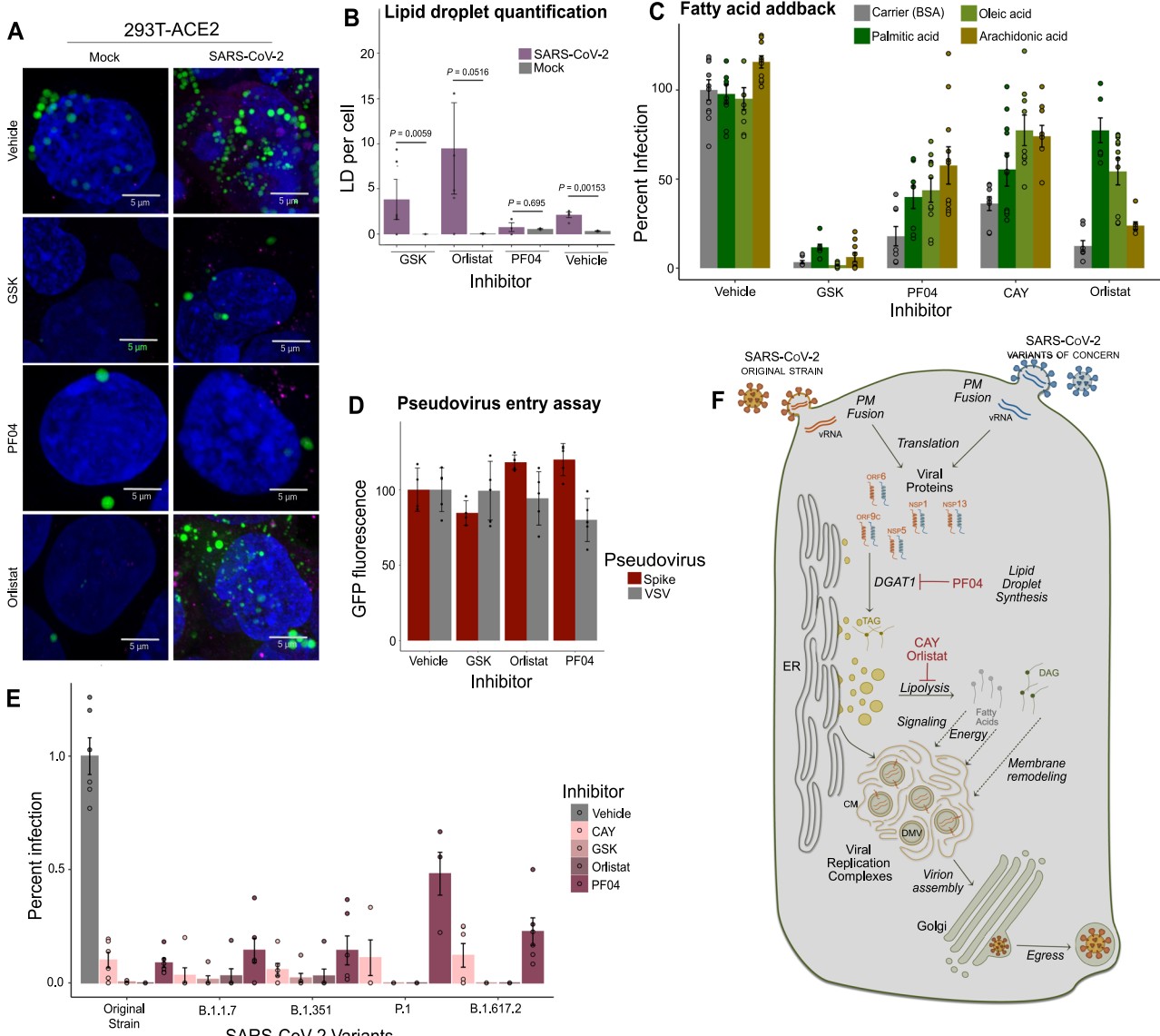

**Fig. 6 Mechanisms and breadth of glycerolipid inhibition against SARS-CoV-2. A** Representative images of HEK293T-ACE2 cells treated with each indicated inhibitor (10 μM) or vehicle (DMSO), infected with SARS-CoV-2 for 48 h (MOI = 1), and stained to visualize lipid droplets (BODIPY 493/503), and dsRNA. Images are representative of three independent experiments. **B** Quantification of lipid droplet numbers in (**A**). Data are mean ± SE for n = 6 biological replicates; P-values are derived from one-way ANOVA. **C** Partial rescue of inhibition of viral replication by supplementation with exogenous fatty acids. Inhibitors and fatty acids are both used at 10 μM, administered simultaneously overnight before a 48 h infection (MOI = 0.1). Data are mean ± SE for n = 10 biological replicates, from three independent experiments. **D** GFP fluorescence resulting from an infection with lentivirus pseudotyped with either SARS-CoV-2 Spike protein or VSV G protein. Data are mean ± SD for n = 5 biological replicates. **E** Inhibition of the original strain and four variants of concern of SARS-CoV-2 in 293T-ACE2 cells by four inhibitors of glycerolipid biosynthesis, each at 10 μM overnight prior to an 48-h infection (MOI = 0.1). Data are from three independent experiments; data are mean ± SE for n = 6 biological replicates. **F** A model for neutral lipid flux during SARS-CoV-2 infection. vRNA viral RNA; PM plasma membrane; DGAT1 diacylglycerolacetyltransferase 1; TAG triacylglycerol; DAG diacylglycerol; ER endoplasmic reticulum; CM convoluted membrane; DMV double-membraned vesicle Source data are provided as a Source Data file.

Since we find that lipolysis, TAG synthesis, and de novo fatty acid synthesis are all critical for viral replication, we went on to ask whether inhibition of any of these enzymes could be overcome by the addition of exogenous fatty acids. We simultaneously treated HEK293T-ACE2 cells with each inhibitor, as above, and simultaneously with BSA precomplexed with 10 μM palmitic acid, oleic acid, arachidonic acid, or fatty acid-free BSA carrier alone. After this overnight treatment, we infected with SARS-CoV-2, and harvested supernatants after 48 h to measure released virus. We found that while the BSA carrier alone did not affect viral production or the action of any of the inhibitors, palmitic acid was able to at least partly, and in some cases almost

completely, restore SARS-CoV-2 infection to vehicle-treated levels (Fig. 6C). This was particularly true for the inhibitors of lipolysis, CAY10499 and Orlistat, highlighting the critical role of lipolysis in the viral life cycle. Furthermore, for CAY10499, which inhibits lipolysis but not fatty acid synthesis, the polyunsaturated fatty acid arachidonic acid more potently rescued infection than the saturated fatty acid palmitic acid.

**Lipid phenotypes conserved across variants of concern**. The continued global transmission of SARS-CoV-2 has led to the emergence of variants of concern (VOC) that show evidence of

increased transmissibility[50] or resistance to prior immunity[51,52]. The major VOCs include the B.1.1.7 (also called the Alpha variant), first identified in southeast England in November 2020[53]; B.1.351 (Beta variant), identified in November 2020 in South Africa[54]; P.1 (Gamma variant), identified in December 2020 in Brazil[55]; and B.1.617.2 (Delta variant), identified in October 2020 in India[56]. Several recent studies have shown that these strains escape neutralization of serum antibodies collected from individuals that received COVID-19 vaccine or were previously infected. Most of the mutations in the emerging VOCs are on the spike protein, and while there are some reported alterations in nonstructural proteins, mutations that fundamentally perturb the virus's ability to manipulate host pathways likely come with a quite high fitness cost. We hypothesized, therefore, that the replication of the variants of SARS-CoV-2 is inhibited to a similar degree to the original USA/WA1/2020 strain.

To test if the small molecules that inhibit glycerolipid biosynthetic machinery are broadly efficacious, we used the Alpha, Beta, Gamma, and Delta strains, as well as the WA1 original strain, to infect cells that had been pre-treated overnight with 10 μM CAY10499, GSK2194069, PF04620110, and Orlistat, and assessed viral proliferation by focus-forming assay. We performed these experiments in both HEK293T-ACE2 cells and Caco2 cells. We observed only minor differences in the efficacy of the compounds among the four strains tested (Fig. 6E and Supplementary Fig. 6). GSK2194069 and Orlistat comprehensively block infection (< 5% of vehicle treatment) in both cell types and all five strains. CAY10499 has slightly different efficacies between the two cell lines, (~ 5-10% infection in HEK293T-ACE2, ~ 30% infection in Caco2), but there is no statistical difference between the variants within each cell line. PF04620110 resembles CAY10499 in Caco2 cells; in HEK293T cells, PF04620110 shows reduced efficacy against the P.1 strain. In the delta strain, CAY10499 showed a slightly significant reduction in foci in Caco2 cells ($P = 0.045$), from ~ 30% infection in WA to ~ 5% infection in delta; no other inhibitors were significantly different. Overall, these results show an encouraging conservation of inhibitor efficacy against the four variants of concern in two cell lines.

## Discussion

Based on our integrated lipidomics, microscopy and small-molecule inhibition experiments, we propose here a model for how SARS-CoV-2 uses lipid droplets to support infection (Fig. 6F). We show that host cells undergo profound lipid remodeling after infection with SARS-CoV-2, and that many distinctive features of infection are conserved in infection across multiple cell types. We show that lipid droplet proliferation is a consequence of infection, and that both TAG synthesis and lipolysis are required for effective replication. The lipid droplet phenotype appears to be part of a profound reprogramming of cellular lipid metabolism which is induced directly by individual viral proteins. We find that many additional lipid species and families change upon infection, the investigation of which is far beyond the scope of this paper. Notably, ceramide increases (especially dramatically in A549-ACE2 cells), in agreement with previous observations in Vero E6 cells[57], a lipid with profound structural and signaling roles. Lysoglycerolipids generally decrease (more in HEK293T-ACE2 cells than A549-ACE2 cells). Lysolipids have known signaling functions and also impact membrane structure;[58] their decrease may suggest that they are not being generated—that the infected cell has a greater requirement for the diacyl structure. Decreases in LPCs and PCs have been observed in sepsis[59,60], cancer[61], dengue[62], and Ebola[63] infection; however, the mechanisms behind these phenotypes are

poorly understood. In general, polyunsaturated glycerolipids lipids are dramatically increased while saturated lipids are decreased, suggesting that viral membrane structures require a particularly high level of fluidity.

Many intracellular pathogens are known to hijack lipid droplets in a variety of ways to support their life cycles. Hepatitis C virus (HCV) uses lipid droplets as the site of viral assembly[36], and lipid droplet accumulation as part of hepatic steatosis is characteristic of the disease[37]. Rotaviruses also cause lipid droplets to increase upon infection, and replicate in cytoplasmic inclusions which colocalize with lipid droplets[64]. Dengue virus, meanwhile, consumes host lipid droplets and appears to use them as a source of fatty acids for beta-oxidation[38], while replicating in distinct compartments more closely associated with the ER. In the case of SARS-CoV-2, we show that lipid droplets increase after infection, and that they do not colocalize with sites of viral replication. This suggests that SARS-CoV-2 requires lipid droplets not as a platform for replication, but for their roles in buffering lipid levels and facilitating membrane plasticity to support the ambitious coronaviral membrane rearrangements. This result is in agreement with our finding that purely accumulating lipid droplets is not strictly beneficial for viral replication: the lipolysis inhibitor Orlistat, while allowing for dramatic proliferation of lipid droplets, is still a potent inhibitor of viral replication.

To date, several studies have been published which attempt to correlate metabolite changes in patient plasma either with disease status or disease severity. Such studies capture information about circulating, extracellular lipid species, not the subcellular type of change observed in this study; however, there are some recurring themes which overlap with some of the observations we make here. An increase in TAG is reportedly associated with disease status[8] or severe disease[10,65], as are unsaturated fatty acids[11,66], and a general decrease in serum lysolipids after infection was reported in one study[67]. This suggests that the changes in subcellular lipid composition are indeed key aspects of the virus's pathogenicity, although more studies will be necessary to ascertain whether increased levels of TAG and PUFAs in serum are a marker of disease—an indication of successful intracellular replication—or exerting damage in their own right.

To ask whether SARS-CoV-2 fundamentally requires host lipid metabolic pathways for its survival and proliferation, we challenged the virus with small-molecule inhibitors of glycerolipid metabolism. Our findings highlight the dynamic and specific involvement of host lipids in infection: SARS-CoV-2 requires both de novo fatty acid and TAG synthesis, and lipolysis, simultaneously promoting lipid synthesis and providing specific lipids for viral processes. We further showed that these inhibitors work as effectively against the recently emerging SARS-CoV-2 variants of concern as they do against the original WA1 strain, demonstrating the advantage of designing host-targeted therapeutics against a conserved host dependency pathway.

Our findings fill an important gap in our understanding of host dependency factors of coronavirus infection. Our systematic analysis of the protein-by-protein effect on host lipids reveals a complex network of many individual viral proteins responsible for diverse aspects of host lipid remodeling. Both of our lipidomics datasets are resources for understanding cellular disease pathology and suggest potential directions for therapeutic discovery, highlighted by the success of several inhibitors of glycerolipid biosynthesis in blocking viral replication. In light of the evolving nature of SARS-CoV-2, it is critical that we understand the basic biology of its life cycle in order to illuminate additional avenues for protection and therapy against this global pandemic pathogen, which spreads quickly and mutates with ease.

## Methods

**Cell lines**. Cell lines (HEK293T, HEK293T-ACE2, Vero-E6, A549-ACE2, and Caco2) were obtained from ATCC.

**Viral strains**. SARS-CoV-2 viral strains (isolate USA-WA1/2020: Identifier #NR-52281; isolate USA/CA_CDC_5574/2020: Identifier #NR-54011; isolate hCoV-19/South Africa/KRISP-K005325/2020: Identifier #NR-54009; hCoV-19/Japan/TY7-503/2021: Identifier #NR54982; isolate hCoV-19/USA/PHC658/2021: Identifier # NR-55611) were obtained from BEI resources and propagated in Vero E6 cells.

**Recombinant DNA**. Plasmids containing strep-tagged SARS-CoV-2 proteins were obtained from the Krogon lab at UCSF[59].

**Chemicals and antibodies**. Inhibitors of lipid biosynthesis were obtained from Cayman Chemical; EquiSPLASH lipidomics internal standard was obtained from Avanti Polar Lipids. Anti-dsRNA antibody was obtained from Millipore (identifier MABE1134); anti-mouse IgG AlexaFluor 647 was obtained from Invitrogen (Identifier A32628); anti-llama secondary HRP, goat IgG was obtained from Novus (identifier NB7242). 5 mM BSA-palmitate complex was obtained from Cayman Chemical (identifier 29558); fatty acid-free BSA was obtained from Thermo Fisher (identifier BP9704100).

**Cell culture**. Unless otherwise stated, cells were maintained at all times in standard tissue-culture-treated vessels in DMEM supplemented with 1% non-essential amino acids and 1% penicillin-streptomycin at 37 °C and 5% CO2. Media for Vero-E6 cells, 293 T (wt) and 293T-ACE2 cells was supplemented with 10% FBS while media for Caco2 cells was supplemented with 20% FBS. A549-ACE2 cells were maintained in F12-K media supplemented with 10% FBS and 1% penicillin-streptomycin and 1% non-essential amino acids.

**SARS-CoV-2 growth and titration**. All SARS-CoV-2 isolates were obtained from BEI resources: USA/WA1/2020 (NR-52281), USA/CA CDC 5574/2020 [lineage B.1.1.7] (NR-54011), hCoV-19/South Africa/KRISP-K005325/2020 [lineage B.1.351] (NR-54009), hCoV-19/Japan/TY7-503/2021 [linage P.1] (NR-54982), hCoV-19/USA/PHC658/2021 [lineage B.1.617.2] (NR-55611). Unless otherwise stated, infection assays were performed with USA-WA1/2020. To propagate each virus strain, sub-confluent monolayers of Vero E6 cells were inoculated with the clinical isolates (MOI < 0.01) and grown for 72 h, at which time significant cytopathic effect was observed for all strains. Culture supernatants were removed, centrifuged 10 min at $1000 \times g$, and stored in aliquots at −80 °C. To determine titer, focus-forming assays were performed on the culture supernatant (assay described in detail below). Substantial differences were noted in the focus phenotypes of these five strains.

**Lipidomics—infection**. 293T-ACE2 or A549-ACE2 cells were seeded to 70% cell density (this represented about $4 \times 10^6$ cells per 10 cm dish for 293T-ACE2 cells, and about $1.5 \times 10^6$ cells per 10 cm dish for A549-ACE2 cells). Cells were then inoculated with USA-WA1/2020 (MOI = 5) for 1 h at 37 °C in 2% FBS DMEM, rocking gently every 15 min. After 1 h, infection media was removed and replaced with normal 10% DMEM. Cellular lipids were extracted 24 h after infection. Five biological replicates were infected for 293T-ACE2 cells; nine biological replicates were infected for 549-ACE2 cells.

**Lipidomics—transfection**. Plasmids containing Strep-tagged viral proteins were generously provided by the Krogan lab at UCSF, and have been described previously[68]. Wild-type 293T cells were seeded in 6 cm dishes and transfected with varying amounts viral plasmids (based on optimal expression for each plasmid, see Table S1), as well as a PLVX empty vector control, using Lipofectamine 3000 (ThermoFisher Scientific) as per manufacturer's instructions. Transfection media was carefully removed 6 h after addition and replaced with DMEM. Each condition was repeated in biological quintuplicate. Cellular lipids were extracted 48 h after transfection.

**Lipidomics—lipid extraction**. Cells were washed with PBS and resuspended in a 2:1: 0.75 mixture of chloroform: methanol: water, and 10 μL of an internal standard cocktail (Avanti EquiSPLASH) was added. Extracts were left for 1 h at 4 °C, then the layers were separated by centrifugation ($3000 \times g$ for 10 min), and the chloroform layer was moved to a fresh tube. 2 mL fresh chloroform was added to the aqueous layer, mixed, left for 1 h at 4 °C, separated by centrifugation, and then added to the first chloroform layer. The combined chloroform layers were dried under a stream of nitrogen. These dried extracts were frozen at −80 °C and sent to PNNL on dry ice.

**Lipidomics — LC-MS/MS analysis and lipid identification**. LC-MS/MS parameters were established and identifications were conducted as previously described[69]. A Waters Aquity UPLS H class system interfaced with a Velos-ETD Orbitrap mass spectrometer was used for LC-ESI-MS/MS analyses. Briefly, lipid

extracts were dried under vacuum, dissolved in a solution of 10 μL chloroform plus 540 μL of methanol, and 10 μL were injected onto a reverse-phase Waters CSH column (3.0 mm×150 mm x 1.7 μm particle size), and lipids were separated over a 34-min gradient (mobile phase A: ACN/H2O (40:60) containing 10 mM ammonium acetate; mobile phase B: ACN/IPA (10:90) containing 10 mM ammonium acetate) at a flow rate of 250 μL/min. Samples were analyzed in both positive and negative mode, using higher-energy collision dissociation and collision-induced dissociation to induce fragmentation. Lipid identifications were made using previously outlined fragment ions[69]. The LC-MS/MS raw data files were analyzed using LIQUID[69], and then all identifications were manually validated by examining the fragmentation spectra for diagnostic and fragment ions corresponding to lipid acyl chains. Identifications were further validated by examining the precursor ion isotopic profile and mass measurement error, extracted ion chromatogram, and retention time for each identified lipid species. To facilitate quantification of lipids, a reference database for lipids identified from the MS/MS data was created, and features from each analysis were then aligned to the reference database based on their m/z, and retention time using MZmine 2[70]. Aligned features were manually verified, and peak apex-intensity values were reported for statistical analysis.

**Lipidomics—QC, normalization, and statistical comparison methods**. Lipidomics data were collected in positive and negative ionization mode and analyzed using R. Each ionization mode datasets was normalized using an IS specific to the respective ionization mode. We required that an IS be quantified for every sample to be considered for normalization purposes. Further, normalization factors should not be related to the biological groups being compared to avoid the potential introduction of bias into the data. Thus, for each ionization mode, we evaluated all IS normalization candidates and (1) conducted a test for a difference in mean normalization factors (IS values) by group (Mock vs Virus) and (2) calculated the coefficient of variation (CV) of IS values. The IS showing no evidence of a difference in values by group ($P$-value > 0.5) and with the minimum CV was selected for normalization. The IS '15:0-18:1(d7) PC_IS' was selected based on the above criteria for both positive and negative ionization data and was used as the normalization factor (log2(abundance/IS abundance)) in both datasets, with a mean CV if 25.8% over the two ionization mode datasets. A one-way analysis of variance (ANOVA) was run on each lipid. The resulting $P$-values were adjusted for multiple comparisons within each lipid using the Benjamini-Hochberg multiple test correction[71].

**Lipid droplet immunofluorescence—infection**. 293T-ACE2 or Caco2 cells were seeded at 70% cell density in 24-well plates and allowed to grow overnight. Cells were then inoculated with USA-WA1/2020 (MOI = 1) for 1 h at 37 °C in 2% FBS DMEM, rocking gently every 15 min. After 1 h, infection media was removed and replaced with normal 10% DMEM (or 20% DMEM, for Caco2 cells). Cells were fixed 8 h, 24 h, or 48 h after infection in 4% PFA.

**Lipid droplet immunofluorescence—transfection**. Wild-type HEK293T cells were seeded in glass-bottomed 24-well plates at 70% cell density and transfected with varying amounts viral plasmids (based on optimal expression for each plasmid, see Table S1), as well as a pLVX empty vector control, using Lipofectamine 3000 (ThermoFisher Scientific) as per manufacturer's instructions. Transfection media was carefully removed 6 h after addition and replaced with DMEM. Cells were fixed 48 h after transfection in 4% PFA.

**Lipid droplet immunofluorescence—imaging**. After fixation, cells were washed three times with PBS, permeabilized with 0.01% digitonin in PBS for 30 min, and blocked with 5% Normal Goat Serum in PBS. Cells were stained overnight with an anti-dsRNA antibody diluted 1:50 in blocking buffer (infections), or an anti-Strep antibody diluted 1:250 in blocking buffer (transfections). Cells were washed three times with PBS and then stained with an A647 secondary antibody (1:500) for 1 h. Cells were then stained with 1 μg/mL BODIPY 493/503 in PBS for 15 min, and then 1x DAPI for 10 min. Cells were imaged on a Zeiss LSM 980 Laser-Scanning 3-channel confocal microscope with Airyscan.2.

**Lipid droplet immunofluorescence—image analysis**. Pearson's correlation coefficients were measured from 2D projections of z-stacks in Cellprofiler 3.1.8.[72]. Lipid droplets were counted and their sizes, in number of pixels, were measured, using a Cellprofiler pipeline. For transfection experiment, regions of interest were first identified based on intensity in the Strep channel so that only transfected cells were analyzed.

**Cytotoxicity screening**. 293T-ACE2 and Caco2 cells were seeded in 96-well plates. The next day they were treated with six 5-fold dilutions of each compound, starting from 50 μM. Each condition was tested in triplicate. After 72 h of compound treatment, cytotoxicity was assessed using resazurin, which is converted into fluorescent resarufin by cells with active oxidative metabolism[40]. Resazurin was added to a concentration of 0.15 mg/mL and cells were left at 37 °C for 4 h, and then fluorescence intensity was measured using a BMG CLARIOstar fluorescence plate reader with 560 nm excitation/590 nm emission.

**Single concentration screen for replication inhibition (all strains of SARS-CoV-2)**. The highest concentration for each inhibitor that did not cause cytotoxicity was selected for this assay. For most described inhibitors 10 µM was used, except remdesivir (2 µM). Each cell line (Caco2 or 293T-ACE2) was seeded in 96-well plates at a density of 10,000 cells per well and treated overnight with each inhibitor prior to infection with SARS-CoV-2 with an MOI of 0.1. The infection was continued for 48 h. To quantify viral production, focus-forming assays were performed on the supernatants, described in detail below.

**Pseudovirus lentivirus production**. 293 T cells were seeded at 2 million cells/dish in 6 cm TC-treated dishes. The following day, cells were transfected as described above with lentivirus packaging plasmids, SARS-CoV-2 S plasmid, and IzGreen reporter plasmid[73]. After transfection, cells were incubated at 37 °C for 60 h. Viral media was harvested, filtered with a 0.45 µm filter, then frozen before use. Virus transduction capability was then determined by fluorescence using a BZ-X700 all-in-one fluorescent microscope (Keyence), and a 1:16 dilution of viral stocks was found to be optimal for neutralization assays.

**Pseudovirus entry assay**. Neutralization protocol was based on previously reported experiments with the SARS-CoV-2 S pseudotyped lentivirus[73]. 293T-ACE2 cells were seeded on tissue-culture-treated, poly-lysine treated 96-well plates at a density of 10,000 cells per well. Cells were allowed to grow overnight at 37 °C, and then treated with selected inhibitors as described above for live virus infection. Lz-Green SARS-CoV-2 S pseudotyped lentivirus was added to 293T-ACE2 cells treated with 5 µg/mL polybrene and incubated for 48 h before imaging. Cells were fixed with 4% PFA for 1 h at room temperature, incubated with DAPI for 10 min at room temperature, and imaged with BZ-X700 all-in-one fluorescent microscope (Keyence). The estimated area of DAPI and GFP fluorescent pixels was calculated with built-in BZ-X software (Keyence). There were five biological replicates for each condition, and the biggest outlier was removed from analysis due to inherent variability in the assay.

**Measurement of compound EC$_{50}$**. Compounds from the single concentration screen that showed efficacy against SARS-CoV-2 replication were tested to measure compound EC$_{50}$. The cell line of interest (293T-ACE2 or Caco2) was seeded in 96-well plates at a density of 10,000 cells per well, and treated overnight with 2-fold dilutions of each compound, starting from 50 µM for Atglistatin, PF04620110, GSK2194069, and CAY10499, and starting at 1 µM for Orlistat. Each condition was tested in quadruplicate. The next day cells were infected as described above, and the infection was continued for 48 h, and then the supernatants were used in a focus-forming assay, as described below.

**Fatty acid supplementation to inhibitor-treated cells**. HEK293T-ACE2 cells were seeded and treated overnight with inhibitors as above ("single concentration screen"), and simultaneously treated either with 10 µM of a BSA-palmitate complex or 10 µM of a corresponding fatty acid-free BSA solution. Dishes were infected with SARS-CoV-2 (WA1 strain, MOI = 0.1), and supernatants were collected for focus-forming assay 48 h after infection.

**Focus-forming assay**. Vero E6 cells were seeded in a 96-well plate at a density of 20,000 cells per well. The next day, supernatants from infected Caco2 or 293T-ACE2 cells were diluted by adding 225 µL dilution media (Opti-MEM, 2% FBS, 1% pen-strep, 1% non-essential amino acids) to a U-bottom 96-well plate, and then 25 µL of virus-infected supernatant. Further dilutions were made in the same manner, if so desired. Media from the Vero E6 cells was removed and 25 µL diluted virus was added to each well. Vero E6 cells were inoculated for 1 h at 37 °C/5% CO2 with occasional rocking. After 1 h, 125 µL of overlay media (0.01 mg/mL methylcellulose in dilution media) was added to each well. Plates were incubated at 37 °C for 24 h. Overlay media was removed, and replaced with 4% PFA. Plate and lid were saturated in 4% PFA for at least 1 h at room temperature and removed from the BSL-3. PFA was washed off by gently immersing the plate in a vat of deionized water. Plates were permeabilized in perm buffer (0.1% saponin, 0.1% BSA in PBS) for 30 min, then incubated with 50 µL primary antibody (alpaca anti-SARS-CoV-2 serum, diluted 1:5,000 in perm buffer) for either 2 h room temperature or overnight at 4 °C. Antibody was removed and plates were washed 3 × 5 min with 200 µL/well PBST (0.1% tween in PBS). Plates were incubated with 50 µL secondary antibody (anti-llama HRP, goat IgG) for either 2 h room temperature or overnight at 4 °C. Antibody was removed and plates were washed 3 × 5 min with 200 µL/well PBST. Plates were stained with 50 µL/well TrueBlue peroxidase substrate for 30 min. Foci were imaged on an ImmunoSpot S6 Macro ELISPOT imager, and then counted using the Viridot R package[74].

**Quantification and statistical analysis**. EC$_{50}$ values were calculated using the Hill equation. Unless otherwise stated, $P$ values are from one-way ANOVA tests without adjustments for multiple comparisons, with $P < 0.05$ considered statistically significant.

**Reporting summary**. Further information on research design is available in the Nature Research Reporting Summary linked to this article.

## Data availability

The raw lipidomics datasets generated during this study have been deposited in the MassIVE mass spectrometry database under accession code MSV000087944 (https://doi.org/10.25345/C51Z6R). The processed lipidomics data are provided in Supplementary Data 1 (live virus lipidomics, HEK293T-ACE2 cells), and Supplementary Data 2 (live virus lipidomics, A549-ACE2 cells), and Supplementary Data 3 (viral-protein lipidomics). Source data for other figures are provided for this paper. Source data are provided with this paper.

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

## Acknowledgements

This work was supported by the National Institutes of Health (1RO1AI141549, received by FGT) and the Pulmonary & Critical Care training grant, "Multidisciplinary Research Training in Pulmonary Medicine" (T32HL083808, received by T.A.B.). Lipidomics analyses were performed in the Environmental Molecular Sciences Laboratory, a national scientific user facility sponsored by the Department of Energy (DOE) Office of Biological and Environmental Research located at the Pacific Northwest National Laboratory (PNNL). PNNL is a multiprogram national laboratory operated by Battelle for the DOE under Contract DE-AC05-76RLO 1830. J.E.K. and L.M.B. were supported by Laboratory Directed Research and Development Program at PNNL, and the National Institute of Environmental Health Sciences grant (U2CES030170, received by T.O.M.). PNNL is a multiprogram national laboratory operated by Battelle for the U.S. Department of Energy under contract DE-AC05-76RLO 1830.

## Author contributions

F.G.T., S.E.F., J.E.K., and H.C.L. designed the study; S.E.F., J.E.K., H.C.L., T.A.B., J.B.W., and L.M.B. performed the experiments; S.E.F. wrote the original draft of the manuscript; all authors reviewed and edited the manuscript; S.E.F. and J.-Y.L. visualized the data; F.G.T. supervised and administered the project; F.G.T., C.S., J.E.K., and T.O.M. acquired funding.

## Competing interests

The authors declare no competing interests.
