## [Peer Review File · Nature Communications]

Reviewer comments, first round review -

Reviewer #1 (Remarks to the Author):

Overall this is a very interesting manuscript and technically well-executed. However, there are few points that need to be addressed. Why the lipidomic has been performed on a single cell line HEK293T cells? Would it be more appropriate to use lung cells for example A549-Ace or human H522 lung adenocarcinoma cells?

It would also be interesting to correlate the present findings with lipidomic analysis available in literature such as large-scale plasma analysis. Is there any correlation or similarities in the alteration of the lipid profile? In other words, the lipidomic data are a bit descriptive and more in-depth analysis and interpretation would improve the impact of the manuscript. For instance, alterations in lysophospholipids levels have been reported in other SARS-CoV-2 lipidomic studies. Their potential role and mechanism of production and action should be discussed. Why 24 hours has been used as the time of infection?

Reviewer #2 (Remarks to the Author):

Farley et al., who carried out lipidomics on SARS-CoV2 infected cell lines and identified lipometabolic pathways modulated by COV2 including increases in lipid droplet numbers and/or sizes (although not novel as it was also shown by Dias et al., Plos Pathogens 2020). Authors also identify specific glycerolized synthesis and lipolysis inhibitors that block SARS-CoV2 replication. This is potentially a useful resource but several issues need to be addressed.

Have the authors looked at the impact of individual CoV2 proteins on LD numbers or size?

Its unclear how these drugs are inhibiting the viral replication- It seems to me that they are blocking entry of the virus (Figure 6E)- have the authors done time kinetics of drug addition: adding it before infection, after infection etc.

The legends do not match the figures. Look at Figure 6 for example.

Reviewer #3 (Remarks to the Author):

This manuscript presents interesting data on the lipidomic changes in the host cells from SARS-CoV-2 infection. The study is highly comprehensive with respect to lipidomic analysis and identifies lipid droplet accumulation in the host cells as a result of viral infection. The authors further delineated the effect of specific structural proteins on the viron in eliciting the lipidomic changes. It is interesting to note that lipids containing polyunsaturated fatty acids (PUFA) are enriched in the lipid droplets with a concomitant decrease in the lipids with saturated fatty acids. Further, it appears that triglyceride (TAG) accumulation characterizes the lipid droplet formation, while both TAG synthesis and TAG hydrolysis appears to be important for infection. However, the lipid droplets are not associated with dsRNAs of the virions and hence not clear how this enrichment aids infection. While the authors chose two different cell lines to demonstrate the generality of the changes in host cells, neither of them are of pulmonary origin. Since the primary target of SARS-CoV-2 is lungs, the choice of cell lines diminishes the relevance of the data to COVID-19.

Enrichment of TAGs containing PUFA coupled with decrease in saturated fatty acid containing TAG may indeed suggest increased membrane fluidity as the authors suggested, but no explanation was offered in the absence of association of the lipid droplets with the viron. It is known that further metabolism of PUFA leads to lipid mediators with profound influence on signaling pathways and biology of the cell. Selective enrichment of PUFA and the concomitant decrease of saturated fatty acids suggest that the change is not merely related to energy requirements to assist infection. It is also clear that the dynamic process of fatty acid esterification to TAGs as well as the

hydrolysis by lipases to release the fatty acids is important for infection. This suggests that a selective release of fatty acids to feed into downstream metabolism or further lipid remodeling is in operation. A classic example of such selective release is the supply of arachidonic acid from cell membranes by specific phospholipases A2 to cyclooxygenases (Reddy, ST, Herschman, HR. Prostaglandin synthase-1 and prostaglandin synthase-2 are coupled to distinct phospholipases for the generation of prostaglandin D2 in activated mast cells. J Biol Chem. 1997;272(6):3231-7). While the observations reported are important, addressing the following questions will significantly improve our understanding (in addition to using an appropriate cell line of pulmonary origin):

Are the fatty acids released by lipases characterized?

If PUFA are released, does further metabolism of PUFA play a role?

Can the fatty acids be supplemented to overcome the lipase inhibitor effects? Based on the data on TAG biosynthesis and lipase inhibitors, one would surmise that the release of fatty acids from TAG stores of the lipid droplets play a crucial role. So, to corroborate the TAG synthesis/TAG lipolysis paradigm, identification of the released fatty acids and supplementation to rescue from lipase inhibition should be tested. Otherwise, it is difficult to tease out the off-target effects of these inhibitors.

While the lipidomic analysis is robust, it is not clear why the authors chose negative ion mode for the analysis of ceramides and phosphatidylethanolamines. Both of these lipid sub-classes are detectable with nearly two orders of higher sensitivity in the positive ion mode without sacrificing selectivity or ability to glean structural information from the dependent mass spectrum.

Reviewer #1

Overall this is a very interesting manuscript and technically well-executed. However, there are few points that need to be addressed. Why the lipidomic has been performed on a single cell line HEK293T cells? Would it be more appropriate to use lung cells for example A549-Ace or human H522 lung adenocarcinoma cells?

In the original conception of this paper, we were primarily focusing on the transfection of viral proteins to observe changes induced by each viral protein. In order to observe meaningful results, high transfection efficiency was required, and we selected HEK293T cells for this purpose. The point that a human lung epithelial line would be more relevant to the live virus infection is well taken, however, and is reflected in updated Figure 1 and Figure S1, where we presented the lipidomics of A549-ACE2 cells infected with SARS-CoV-2 virus. While we do see some differences between the lipid remodeling in the two cell lines in response to SARS-CoV-2 infection (which is perhaps less surprising than if the response had been identical), we also see important commonalities, in particular a consistent increase in TAG species after infection, a general shift in phospholipid acyl chains from unsaturated to polyunsaturated, and an increase in ceramide, suggesting to us that these lipid species are fundamental to the intracellular survival and replication of SARS-CoV-2.

It would also be interesting to correlate the present findings with lipidomic analysis available in literature such as large-scale plasma analysis. Is there any correlation or similarities in the alteration of the lipid profile?

While of course a plasma lipidome of an infected person and the subcellular lipidome of an infected cell contain quite different information about the pathogenesis of the virus, there are some interesting similarities between our findings and some investigations of the lipidomes of patient sera; we have added this to our discussion [lines 350-359]

In other words, the lipidomic data are a bit descriptive and more in-depth analysis and interpretation would improve the impact of the manuscript. For instance, alterations in lysophospholipids levels have been reported in other SARS-CoV-2 lipidomic studies. Their potential role and mechanism of production and action should be discussed.

We have included more discussion of lysolipids as well (see discussion lines 328-333); thank you for this important comment.

Why 24 hours has been used as the time of infection?

We have found that 24 hours after infection is a meaningful time point to observe host responses to SARS-CoV-2, since earlier time points may not have established as robust a viral presence within the cell, and later time points are often complicated by high levels of cell death. As a further note, for most of our experiments using small-molecule inhibitors to block infection, we chose 48 hours post infection for our readout timeline, because we found that in measuring viral output, the slightly longer time of infection allowed for a larger dynamic range in viral titers, and thus a more robust analysis. From our perspective, the time of infection largely depended on what we were looking for: 24 hours post infection gave us a good portrait of an infected, but still viable cell, while 48 hours post infection gave us a good portrait of the aftermath of a successful (or not) infection.

Reviewer #2

Farley et al., who carried out lipidomics on SARS-CoV2 infected cell lines and identified lipo-metabolic pathways modulated by COV2 including increases in lipid droplet numbers and/or sizes (although not novel as it was also shown by Dias et al., Plos Pathogens 2020). Authors also identify specific glycer-

olized synthesis and lipolysis inhibitors that block SARS-CoV2 replication. This is potentially a useful resource but several issues need to be addressed.

Have the authors looked at the impact of individual CoV2 proteins on LD numbers or size?

Thank you for this important question. To address this comment, we selected for further investigation viral proteins that we had found to increase triacylglycerol (nsp1, nps5, nsp13, orf6, and orf9c), and compared them to empty vector; these results are now shown in fig 4 I and J. Briefly, we found that four of the five of these proteins directly led to an increase in LD number, suggesting their direct role in promoting lipid droplet synthesis.

Its unclear how these drugs are inhibiting the viral replication- It seems to me that they are blocking entry of the virus (Figure 6E)- have the authors done time kinetics of drug addition: adding it before infection, after infection etc.

This is a point we had considered. To address whether the compounds were preventing viral entry or were acting intracellularly, we used a previously described assay (see reference 61), where a lentiviral vector bearing a GFP and coated in the SARS-CoV-2 Spike protein (pseudovirus) was used to infect drug-treated cells (a schematic for this assay can be found in fig S5A). In this assay, any reduction in viral load (as measured by GFP expression) should only be due to a defect in Spike-mediated entry, since the pseudovirus contains none of the other viral material necessary to establish intracellular infection. Since we see no reduction in viral entry by this assay, we conclude that the glycerolipid inhibitors are acting on a intracellular step of the life cycle, and we feel that this is a more robust and specific experiment than a time-of-addition assay.

The legends do not match the figures. Look at Figure 6 for example.

Thank you for this observation. We have rectified this error.

Reviewer #3

This manuscript presents interesting data on the lipidomic changes in the host cells from SARS-CoV-2 infection. The study is highly comprehensive with respect to lipidomic analysis and identifies lipid droplet accumulation in the host cells as a result of viral infection. The authors further delineated the effect of specific structural proteins on the viron in eliciting the lipidomic changes. It is interesting to note that lipids containing polyunsaturated fatty acids (PUFA) are enriched in the lipid droplets with a concomitant decrease in the lipids with saturated fatty acids. Further, it appears that triglyceride (TAG) accumulation characterizes the lipid droplet formation, while both TAG synthesis and TAG hydrolysis appears to be important for infection. However, the lipid droplets are not associated with dsRNAs of the virions and hence not clear how this enrichment aids infection.

While the authors chose two different cell lines to demonstrate the generality of the changes in host cells, neither of them are of pulmonary origin. Since the primary target of SARS-CoV-2 is lungs, the choice of cell lines diminishes the relevance of the data to COVID-19.

Thank you for this comment - we have repeated the lipidomics analysis in A549-ACE2 (human lung epithelial cell line) to underscore the physiological relevance of this study. (See above response to reviewer #1)

Enrichment of TAGs containing PUFA coupled with decrease in saturated fatty acid containing TAG may indeed suggest increased membrane fluidity as the authors suggested, but no explanation was offered in the absence of association of the lipid droplets with the viron.

This is an important comment, and we have now discussed this observation in more detail in our revised manuscript [lines 336-349].

It is known that further metabolism of PUFA leads to lipid mediators with profound influence on signaling pathways and biology of the cell. Selective enrichment of PUFA and the concomitant decrease of saturated fatty acids suggest that the change is not merely related to energy requirements to assist infection. It is also clear that the dynamic process of fatty acid esterification to TAGs as well as the hydrolysis by lipases to release the fatty acids is important for infection. This suggests that a selective release of fatty acids to feed into downstream metabolism or further lipid remodeling is in operation. A classic example of such selective release is the supply of arachidonic acid from cell membranes by specific phospholipases A2 to cyclooxygenases (Reddy, ST, Herschman, HR. Prostaglandin synthase-1 and prostaglandin synthase-2 are coupled to distinct phospholipases for the generation of prostaglandin D2 in activated mast cells. *J Biol Chem.* 1997;272(6):3231-7). While the observations reported are important, addressing the following questions will significantly improve our understanding (in addition to using an appropriate cell line of pulmonary origin):

Are the fatty acids released by lipases characterized? If PUFA are released, does further metabolism of PUFA play a role?

This is a fascinating question, but unfortunately our lipidomics datasets only allow us to look at a snapshot of what is undoubtedly a complex, dynamic situation. It would be difficult to determine which FA are released by lipases from the lipidomic data. Our inhibitor experiments suggest that TAG synthesis is probably primarily an intermediate step in the viral program of lipid remodeling, however, and the fact that CAY inhibition can be prevented by exogenously adding fatty acids of various chain lengths (see below) suggests that fatty acid release by the activity of lipases is a cellular process that is strictly required for infection.

Can the fatty acids be supplemented to overcome the lipase inhibitor effects? Based on the data on TAG biosynthesis and lipase inhibitors, one would surmise that the release of fatty acids from TAG stores of the lipid droplets play a crucial role. So, to corroborate the TAG synthesis/TAG lipolysis paradigm, identification of the released fatty acids and supplementation to rescue from lipase inhibition should be tested. Otherwise, it is difficult to tease out the off-target effects of these inhibitors.

Thank you for this comment and we have experimentally addressed this question (see Figure 6D). We used palmitic acid, oleic acid, and arachidonic acid to complement the effects of GSK (fatty acid synthase inhibitor), PF04 (DGAT1 inhibitor), CAY (broad-spectrum lipase inhibitor), and Orlistat (inhibitor of both FASN and various lipases). We found that most of these fatty acids were able to at least partially rescue infection; of particular relevance to the discussion of lipolysis are our observations with CAY, where saturated palmitic acid is able to partially restore infection to vehicle-treated levels, while oleic acid and arachidonic acid restore infection nearly completely, underscoring again the importance of unsaturation to the virus's program of lipid remodeling.

While the lipidomic analysis is robust, it is not clear why the authors chose negative ion mode for the analysis of ceramides and phosphatidylethanolamines. Both of these lipid sub-classes are detectable with nearly two orders of higher sensitivity in the positive ion mode without sacrificing selectivity or ability to glean structural information from the dependent mass spectrum.

The authors agree that ceramides and PE lipid ionize in both modes. We have chosen negative ionization as more fragment ions are produced in negative ionization. We agree though the positive ionization mode is also a valid approach. Please see the below figure comparing Cer(d18:1/16:0) identified in both negative and positive ionization mode from the same sample that was used in this study.

For PE lipids, we are able to confidently identify many more lipids in negative mode versus positive mode. We recognize that the diagnostic ion for PE lipids is much more intense in positive mode but the

lack of strong fatty acid associated fragment ions and the high intensity of background fragment ions impedes the confident identification of the associated fatty acids. As I am sure the reviewer knows, the fatty acid associated fragment ions are much stronger in NEG mode and we still detected, although at much lower intensity, the diagnostic fragment ion for PE lipids. Please see the below figure comparing PE(18:0_18:1) identified in both negative and positive ionization mode from the same sample that was used in this study.

Cer(d18:1/16:0)

PE(18:0_18:1)

Gray fragment ions in spectra are ions that do not match the identification

Reviewer comments, second round review -

Reviewer #1 (Remarks to the Author):

The authors have satisfactorily addressed all comments and concerns raised in my previous assessment

Reviewer #2 (Remarks to the Author):

Authors have provided satisfactory responses to queries.

Reviewer #3 (Remarks to the Author):

The authors responses and changes to my comments were appropriate.